# Prevalence of Worldwide Neonatal Calf Diarrhoea Caused by Bovine Rotavirus in Combination with Bovine Coronavirus, *Escherichia coli* K99 and *Cryptosporidium* spp.: A Meta-Analysis

**DOI:** 10.3390/ani11041014

**Published:** 2021-04-03

**Authors:** Michael Brunauer, Franz-Ferdinand Roch, Beate Conrady

**Affiliations:** 1Institute of Food Safety, Food Technology and Veterinary Public Health, University of Veterinary Medicine, 1210 Vienna, Austria; m.brunauer@gmx.net (M.B.); Franz-Ferdinand.Roch@vetmeduni.ac.at (F.-F.R.); 2Department of Veterinary and Animal Sciences, Faculty of Health and Medical Sciences, University of Copenhagen, 1870 Frederiksberg C, Denmark; 3Complexity Science Hub Vienna, 1080 Vienna, Austria

**Keywords:** bovine rotavirus, bovine coronavirus, concurrent-infection, *Cryptosporidium* spp., *Escherichia coli* K99, epidemiology, mixed-infection, systematic review, pathogens

## Abstract

**Simple Summary:**

Three weighted-stratified random-effects meta-analyses were performed to estimate the worldwide neonatal calf diarrhoea prevalence of mixed infections of the causative agents bovine rotavirus (BRV), bovine coronavirus (BCoV), *Escherichia coli* K99 (ETEC) and *Cryptosporidium* spp. (Crypto). The highest worldwide mean pooled prevalence was identified for BRV-Crypto (6.69%; confidence interval (CI): 4.27–9.51), followed by BRV-BCoV (2.84%; CI: 1.78–4.08) and BRV-ETEC (1.64%; CI: 0.76–2.75). In all concurrent infections with BRV, the highest mean prevalence was identified in calves with diarrhoea, in dairy herds and in the age classes of sampled animals between 0–14 days. The prevalence of the BRV-BCoV mixed infection is higher than expected based on the ratio of the occurrence of both individual infections in calves with diarrhoea.

**Abstract:**

Multiple enteropathogens such as bovine rotavirus (BRV), bovine coronavirus (BCoV), *Escherichia coli* K99 (ETEC) and *Cryptosporidium* spp. (Crypto) are the most common causes of calf diarrhoea during the first 30 days of animal age. Three weighted-stratified random-effects meta-analyses were performed to calculate the worldwide prevalence of mixed infections of the causative agents (i.e., BRV-BCoV, BRV-ETEC, BRV-Crypto) and their potential influencing factors. The meta-analysis covered 41 studies (94 sub-studies) in 21 countries that determined the presence or absence of mixed infections in global calf populations. The highest worldwide estimated pooled prevalence was identified for BRV-Crypto (6.69%), followed by BRV-BCoV (2.84%), and BRV-ETEC (1.64%). The chance of detecting BCoV in calves with diarrhoea was 1.83 higher in the presence of BRV compared to calves without BRV, whereby an inhibition effect (odds ratio: 0.77) was determined between BRV and Crypto infections. The diagnostic methods were identified as a significant influencing factor in the detection of all considered mixed infections, while the other analysed factors differed in relation to their effect on prevalence. In contrast to BRV-BCoV, the prevalence of BRV-ETEC and BRV-Crypto mixed infections followed the course of individual ETEC and Crypto prevalence related to the age class of the sampled animals.

## 1. Introduction

Neonatal calf diarrhoea (NCD) is a well-known worldwide disease in the cattle industry which causes substantial economic losses due to high morbidity, mortality, growth retardation and treatment costs, as well as serious long-term consequences such as delayed first calving [1,2,3,4,5,6,7,8,9]. NCD is the most common cause of death in dairy calves during their first 30 days of age with a case fatality risk of approximately 5% [10,11].

Multiple enteric pathogens, viral (e.g., bovine rotavirus, bovine coronavirus, bovine viral diarrhoea virus), parasitic (e.g., *Cryptosporidium parvum*, *Giardia duodenalis*, *Eimeria* spp.) and bacterial (e.g., *Escherichia coli* K99, *Salmonella* spp., *Clostridium perfringens*) are infectious causative agents of NCD [1,12,13,14]. Bovine rotavirus (BRV), bovine coronavirus (BCoV), enterotoxigenic *Escherichia coli* K99 (ETEC) and *Cryptosporidium parvum* are the most frequently identified causative factors of calf diarrhoea during the first 30 days of age [15,16,17,18], whilst BRV is the most commonly involved pathogen in mixed infections [14,19,20]. Neonatal calves are most susceptible to infections with ETEC in the first four days of life [15,21]. In the first to second week of age, infections with BRV are most common, whilst infections with BCoV occur more frequently from days five to 20 [22]. Between the first and the third week, calves are most susceptible to infections with *Cryptosporidium parvum* [23]. To the best of our knowledge, an overview about mixed infections of these pathogens related to animal age across the literature is not available yet.

Concurrent infections of these infectious causative agents are often observed, in particular in calves with diarrhoea compared to healthy calves [17,24,25]. Besides pathogens themselves, there are other factors such as applied diagnostic methods, management-related factors (e.g., dam vaccination, colostral consumption, herd size, biosecurity practice, calf housing, hygienic condition, separation of animal based on age, feeding), and environmental factors (e.g., season of birth) that may influence recording of the occurrence and/or prevalence of enteropathogens [10,26,27,28,29,30].

The objectives of this study were (i) to review the literature systematically regarding the prevalence of BRV infections in combination with BCoV (i.e., BRV-BCoV), ETEC (i.e., BRV-ETEC) and *Cryptosporidium* spp. (Crypto) (i.e., BRV-Crypto; N.B. a differentiation of the species is not possible with formerly commonly used diagnostic methods like acid-fast staining, the analysis here is based on the genus level of *Cryptosporidium* spp. [31]) and potential influencing factors; (ii) to perform weighted random-effects meta-analyses to estimate the overall pooled prevalences across the worldwide studies and to identify sources of heterogeneity of prevalences among the study outcomes (referred as subgroup-analysis), (iii) to determine the statistical influence of potential influencing factors on the reported prevalences of concurrent infections; (iv) to analyse the chance that one of the three pathogens occur in the presence of BRV; (v) to determine the expected prevalence of mixed infection in calves with diarrhoea, assuming that both considered causative agents occur independent from each other and (vi) to model the worldwide prevalence of mixed-infection depending on the age class of sampled animals.

## 2. Materials and Methods

A systematic literature search was conducted to identify studies focusing on the prevalence of mixed infections (i.e., BRV-BCoV, BRV-ETEC, BRV-Crypto). Three online databases were used, considering publications until June 2020: PubMed, Scopus and Web of Science. The following predefined search terms were used to identify the greatest possible number of publications: (neonatal calf diarrhea OR calf diarrhoea OR diarrheic calves OR diarrhoeic calves OR pre-weaned) AND (prevalence) AND (mixed infection OR concurrent infection OR co-infection). Due to the large number of articles returned in Scopus and Web of Science, the search terms were set in quotation marks to ensure that the online databases only return publications with the exact sequence of words. Studies returned by the online databases were defined as ‘primary literature’ and were screened in full by one reviewer (MB) regarding the predefined criteria shown in Table 1 and were reviewed again for validation by one reviewer (FR). Additionally, the reference lists of the primary literature were reviewed regarding article title and abstract for further appropriate studies (MB). Studies from the reference lists were defined as ‘secondary literature’. Uncertainties regarding the inclusion and/or recording of data from the studies were discussed between all authors until a consensus was reached.

The number of identified studies (primary and secondary literature) and the study selection workflow, in accordance with the PRISMA (Preferred Reporting Items for Systematic Reviews and Meta-Analysis) guidelines, are presented in Figure 1. The data collected (Table 1) from the studies (i.e., prevalences of concurrent infections (i.e., BRV-BCoV, BRV-ETEC, BRV-Crypto), occurrence of the individual enteropathogens (BRV, BCoV, ETEC, Crypto), geographical region, sampling period, number of herds, herd type, age of sampled animals, health status, number of tested animals, sample type, genotypes, vaccination status, colostrum intake, diagnostic methods, study type) were entered into a Microsoft Excel datasheet Version 16.16.27 (2016). A study was divided into sub-studies if the study covered differences in e.g., herd type, health status and animal age. Because of the consideration of sub-studies, the total number of publications included in the presented study is thus not identical to the total number of observations. The criteria for study inclusion were (i) focusing on diarrhoea in calves aged ≤ 60 days; (ii) reporting prevalences of BRV-BCoV, BRV-ETEC and/or BRV-Crypto as percentage and/or total number of tested and positively tested calves; (iii) consideration of more than one herd and (iv) only original studies on prevalence data. Although NCD is the most common cause of death in dairy calves during their first 30 days of age, we considered studies with age ranges up to 60 days in the analysis because many of the studies published age ranges including animals older than 30 days. Further, age ranges up to 60 days were considered to get a better impression of the development of mixed infection in both dairy and beef production systems instead of single infection in one production system, and to confirm the knowledge in the literature that NCD most frequently appears in the first 30 days of age. A tested animal corresponds to one sample in the analysed studies. All published mixed infections (i.e., double, triple and quadruple) were considered. For instance, BRV-BCoV-Crypto triple infections were incorporated in the analyses for BRV-BCoV and BRV-Crypto, respectively.

The prevalence of mixed infections with BRV (i.e., BRV-BCoV, BRV-ETEC, BRV-Crypto) were analysed in three weighted-stratified meta-analyses using random effect models. The meta-analyses were used to estimate the worldwide pooled prevalences of the mixed infections in the sampled animals. The prevalences were weighted on the inverse of within-study variance and the variability across the studies, according to the PM (Paule and Mandel) method (Appendix A) [32,33]. For variance-stabilisation of the prevalence data distribution, Freeman–Tukey double arcsine transformation was used [34]. The corresponding back-transformation was conducted according to the approach by Miller (Appendix A) [35]. To validate our approach, we used the REML (restricted maximum likelihood) method instead of the PM, for model fitting, whereas both sub-studies and studies were used simultaneously as random factors. To determine the heterogeneity of the incorporated studies in the meta-analysis, i) the Higgins inverse variance (I^2^) index (i.e., the percentage of total variation across the studies) and ii) the Cochran’s Q-Test (i.e., degree of between study variance, whereby *p* < 0.05 indicated heterogeneity) was calculated. I^2^ greater than 50% indicated substantial heterogeneity between studies (I^2^ lay between 0 and 100%) [36,37]. Both, I^2^ and Cochran’s Q-Test provide no information about the factors which cause the heterogeneity [38]. Thus, a weighted-stratified random-effects meta-analysis (subgroup-analysis) based on the factors in Table 1 was performed in order to identify the possible source of heterogeneity. To avoid imprecise calculation, factors incorporating less than 75% data were excluded from the subgroup-analysis (Table 1). The Egger test and a regression test for funnel plot asymmetry were conducted to identify publication bias. The outliers were identified by performing an influential case diagnostic (i.e., DFFITS (difference in fits) value, covariance ratios, estimates of τ2, Cook’s distances and test statistics for (residual) heterogeneity, see Appendix A) [39,40]. The pooled prevalences for concurrent infections of each study and their weight contribution proportion to the meta-analyses was stratified by the health status of the calves and bounded by 95% confidence intervals (CIs) (Appendix A).

Uni- and multivariate meta-regression analyses were performed based on the approach by Scharnböck et al., 2018 to determine the potential significant influence of factors in Table 1 and their explainable proportion on the variability (R^2^) of prevalences of BRV-BCoV, BRV-ETEC and BRV-Crypto [13]. The final multivariate regression analysis includes only most relevant factors without declining the model-fit accuracy. The most relevant factors were non-correlated (N.B. association between the factors were analysed using Goodman–Kruskal–tau), significant factors from the univariate meta-regression analysis not altering the R^2^ by more than 10% of the full multivariate regression model and provided the lowest Akaike information criteria, corrected for small sample size (AICc). The estimated overall mean prevalences of the concurrent infections from the sub-group meta-analysis were used to model the prevalences of mixed-infection depending on the age with the Loess algorithm (Appendix A). The same approach was applied for the prevalence of each of the four considered pathogens because in contrast to mixed infection, knowledge about the prevalence of the individual four pathogens as a function of age is already known. Thus, if the course of the individual prevalences related to animal age matched the knowledge in the literature, we considered the approach as valid for the mixed-infections.

Additionally, we calculated the expected prevalence of each mixed infection under assumption of independency of both considered causative agents. We investigated whether the expected prevalence of the mixed infections was higher or lower as expected based on the ratio of the occurrence of both individual infections in calves with diarrhoea. In order to analyse the association between two individual pathogens, the OR (odds ratio) based on the absolute frequencies of the detected individual pathogens was used as an effect size for the meta-analysis. This allows us to quantify the OR for one pathogen (i.e., BCoV, ETEC, Crypto) when BRV was present. Both pathogens occur independently, if the OR = 1, while OR >1 or <1 indicated dependency. The meta-analyses were implemented in R (Version 3.4.1 R Foundation for Statistical Computing, Vienna, Austria) using the “metafor” and “GoodmanKruskal” package [41,42].

## 3. Results

In total, 41 (94 sub-studies) from 1293 studies in 21 different countries were included in the meta-analysis (Figure 1). In total, 12,208 animals in approximately 2110 herds were tested for concurrent infections worldwide. The highest worldwide mean pooled prevalence (Table 2, Table 3 and Table 4) was identified for BRV-Crypto (6.69%; CI: 4.27–9.51), followed by BRV-BCoV (2.84%; CI: 1.78–4.08) and BRV-ETEC (1.64%; CI: 0.76–2.75). The regression test for funnel plot asymmetry shows no publication bias (BRV-BCoV: z = 0.41, *p* = 0.67; BRV-ETEC: z = 1.59, *p* = 0.11; BRV-Crypto: z = −0.25, *p* = 0.79), no outliers (Appendix A) and no multicollinearity issues across all mixed infections. The validation of the meta-regression analysis with the restricted maximum likelihood (REML) instead of Paule and Mandel (PM) shows no significant differences in the meta-results, no outliers and no publication bias.

The geographical distribution demonstrated that the majority of BRV-BCoV infections were identified in Europe (4.72%; CI: 2.49–7.45), while the highest prevalences of BRV-ETEC (3.70%; CI: 0.32–9.39) and BRV-Crypto (16.61%; CI: 8.03–27.19) were determined in West Asia. In all concurrent infections with BRV, the highest mean prevalence was identified in calves with diarrhoea, in dairy herds and in the age classes of sampled animals between 0–14 days (Table 2, Table 3 and Table 4). The lowest pooled prevalences of the mixed infections were identified in case-control studies. In contrast to BRV-BCoV, the prevalence of BRV-Crypto increased over time (from 1980: 2.01%; CI: 0.00–11.65 to 2011–2019: 9.07%; CI: 4.72–14.44). In this context, the highest pooled prevalence was identified for the more recent diagnostic methods such as lateral flow immunochromatographic assay (BRV-Crypto; RA: 13.49%; CI: 6.80–21.74) in contrast to methods frequently applied in the past such as acid-fast staining (BRV-Crypto; MS: 3.44%; CI: 0.85–7.16; Table 4). Diagnostic methods were identified as a significant influencing factor in the uni- and/or multivariate-meta-regression analyses over all considered mixed infections. The significance and explained variance of the remaining factors on the worldwide prevalences differ between the concurrent infections and is shown in Table 5.

Our study results confirm that the most concurrent infections occur in dairy and beef production systems in an age range up to 30 days (see Table 2, Table 3 and Table 4). The highest mean prevalence of BRV-BCoV (BRV-ETEC and BRV-Crypto) was identified in animals aged between 7–14 days under consideration the sample size (BRV-ETEC: 0–7 days and BRV-Crypto: 7–14 days). Figure 2 shows that in contrast to BRV-BCoV, the prevalence of BRV-ETEC and BRV-Crypto mixed infections follow the course of the individual ETEC and Crypto prevalence related to the age class of sampled animals. The prevalence of the BRV-BCoV mixed infection is higher than expected based on the ratio of the occurrence of both individual infections in calves with diarrhoea (Figure 3). The chance/odds ratio (OR) to detect BCoV in calves was 1.83 (CI: 1.48–2.27) times higher in the presence of BRV compared to calves without BRV, whereby an opposite effect was identified for BRV-Crypto infections (OR 0.77; CI: 0.60–0.99).

## 4. Discussion

To assess studies with specific focus on NCD prevalences caused by BRV in combination with BCoV, ETEC and Crypto, we reviewed 1193 studies in full, of which 41 studies were incorporated in the meta-analysis presented here. BRV was used as reference for the comparison because it is the most common infectious causative agent in combination with other pathogens of NCD [14,19,20]. As far as the authors are aware, this is the first worldwide meta-analysis to be carried out regarding the concurrent infections of NCD. In contrast to other systematic reviews and meta-analyses with similar focus, our study focused on calves in the most vulnerable age class and took into account the interaction of several pathogens instead of pathogens tested individually [44,45,46].

The results presented here revealed a wide variation in the prevalence of considered mixed infections and their significant influencing factors. The considered causative agents in the presented study cover three (i.e., viruses, bacteria, parasites) of five classes of pathogens in different combinations which differ in their pathogenicity, virulence, infectivity and environmental resistance [47]. This might explain the heterogeneous distribution of the prevalences as well as why the factors differ regarding their significant influence and explained variance on the worldwide prevalences (Table 5).

It is only useful to a limited extent to discuss in detail specific factors on the level of prevalence. For instance, the factor “geographical region” covered several country-specific factors such as average herd size, general law standards, typical husbandry systems, trading systems. All these factors might have a direct or indirect effect on the biosecurity level on farms. For example, Sahlström and colleagues showed that larger farms tend to have higher levels of biosecurity [48], resulting in potentially higher prevalence of infections for areas with smaller farm sizes on average. We assume that this effect might explain the higher prevalence of BRV-BCoV (4.72%; CI: 2.49–7.45) and BRV-Crypto (8.90%; CI: 4.98–13.65) in Europe with smaller structured holdings [49] compared to other regions (Table 2 and Table 4). However, it has been described that factors which influence biosecurity, such as herd size, can also have a direct influence on the incidence of infection [50]. As an example, an accurate uptake of colostrum reduces infections with ETEC [51]. Barry and colleagues showed that calves in smaller herds tend to have higher immunoglobulin G levels [52] which might be a consequence of better colostrum management and/or quality [53].

The results of the meta-analysis presented here confirm the results of several studies [17,24,25] that mixed infections are more common in calves with diarrhoea (BRV-BCoV: 4.22%; BRV-ETEC: 2.26%; BRV-Crypto: 9.41%) than in healthy calves (BRV-BCoV: 0.00%; BRV-ETEC: 0.13%; BRV-Crypto: 0.00%). The lowest prevalences were found across all mixed infections in case-control studies compared to case studies. This can be explained by the fact that as well as calves with diarrhoea, healthy animals were also included. In contrast to BRV-BCoV and BRV-ETEC, an increase in BRV-Crypto prevalence was identified during the period. This can primarily be explained by the use of more sensitive diagnostic methods from 2011 onwards. For instance, the use of microscopy (MS) for Crypto detection (as a single detection method) was mainly found in the studies dated before 1991 and is less sensitive than other diagnostic methods (see diagnostic factor: Several in Table 2, Table 3 and Table 4 and Table 1) used since then. Several authors also reported increasing Crypto prevalence due to more sensitive diagnostic methods [45,54]. For example, the prevalence determined with RA was approximately four times higher than that of MS (Table 4).

The uni- and multivariate regression analysis revealed that the factor “diagnostic method” had a significant impact on the detected prevalence of BRV-BCoV, BRV-ETEC, BRV-Crypto. Although the collected factors in the study presented here can explain a high variance of BRV-BCoV prevalences (R^2^ = 61.23%), it is much less appropriate in the case of BRV-ETEC (R^2^ = 47.83) and BRV-Crypto (R^2^ = 46.20%), which might indicate the presence of other essential factors which were not considered in this study presented here due to the lack of reporting in the literature. These could include factors such as (i) vaccination status of the dam, because colostrum of immunized dams could increase the antibody titre of calves against BRV, BCoV, ETEC in the first month of the animal life [30]. Thus, colostral consumption can decrease the neonatal diarrhoea prevalence, and also reduce shedding of *Cryptosporidium parvum* [55,56]. Information about colostral consumption of calves and vaccination status of dams were specified in 4 (9.76%) and 11 studies (23.83%) respectively. Both factors were not incorporated in the meta-analysis due to the low number of studies (Table 1); (ii) season of sampling, because calves born in the winter season have higher risk of diarrhoea [29] due to lower colostrum quality of the dam [57] and a higher shedding of pathogens (e.g., Cryptosporidium oocysts) throughout the winter season compared to summer [58,59]. Further factors which might influence the prevalences are (iii) e.g., feeding, animal stock intensity and regulations to protect calves [60]. As already mentioned, some of these factors could be indirectly included in the factor “geographical region”. One of the influencing factors that was also not considered in the study presented here is the role of the gut microbiome during pathogenesis at the site of infection in the early life of an animal, and the host-microbial interactions with dietary interventions. A number of studies have analysed the effect of the microbiome composition on new-born health such as on the calf gastrointestinal tract [61,62,63,64,65]. A limitation of our meta-analysis is that the reported prevalences in the studies were not corrected for the varying levels of sensitivity and specificity of the diagnostic tests used (also referred to as apparent prevalence). N.B. only nine studies (21.95%) provided information on the sensitivity and specificity of the applied diagnostic methods (Table 1). Consequently, the worldwide estimated prevalences could be under- and/or overestimated in the presented meta-analysis. Additionally, the estimated worldwide prevalences could be under- and/or overestimated due to our predefined exclusion criteria and/or because studies may not have been identified by the chosen database, search terms and language restrictions. Furthermore, the relatively small number of studies per factor does not allow us to take into account the interaction between the factors. Such interaction would be essential to interpret the results of the subgroup analysis more accurately. For instance, the main reason why the implementation of the antibody-based methods is not reflected in the level of BRV-BCoV infections (Table 2) might be explained by the increasing number of case-control studies since 2001 and thus would explain the decrease in BRV-BCoV prevalence from this year onwards. In general, the results of the subgroup analysis should be interpreted with caution concerning the sample size, the number of studies and broad definition of subgroup factors. The broad definition of factors was used to avoid an unbalanced number of studies per analysed factor. For instance, instead of a range or mean of animal age, it would be more appropriate for epidemiological prevalence studies to summarise the age groups at intervals of seven days and, especially in the first week of age, on a daily basis. This suggestion is supported by other studies [15,21] reporting for example that ETEC frequently occurs in the first four days of animal age. The ability of the *Escherichia coli* K99 antigen to bind on the mucous membrane of the small intestine is age-dependent and gradually decreases from 12 h of animal age [66]. The latter might also explain the course of the prevalence level illustrated in Figure 2. An increase of ETEC after the 3rd week of animal life was observed in the study presented here, a result which was also reported by Izzo and colleagues [67]. This could be related to an immunological gap caused by the decrease of maternal antibodies, while the antibody protection of the calf is not yet sufficient [68]. This decrease in maternal antibodies could also explain the increase of BCoV prevalences in the third week of animal life. Figure 2 shows that BRV prevalence peaks in the first age class of sampled animals, which is a consequence of the short incubation period of 24 h of BRV in combination with a higher susceptibility in this age class [22].

The course of the prevalences of the individual pathogens related to the age class of the sampled animals presented in this study is in accordance with several studies, testing the age dependencies of prevalences of these pathogens [21,23,69,70]. Figure 2 shows that in contrast to BRV-BCoV, the BRV-ETEC and BRV-Crypto mixed infection follows the course of the individual ETEC and Crypto prevalences related to the age class of sampled animals. A prolonged susceptibility and a synergistic interaction between BRV-ETEC has been proven experimentally [71,72,73,74,75]. This observation could be explained by taking into account the fact that rotavirus infection induces important changes in the cytoskeleton which correlate with a decrease in apical expression of disaccharidase [76]. This reduced disaccharidase activity on the cell surface, regardless of whether there is cell damage or not [77], could encourage the growth of bacteria, as described in several studies [15,78]. This synergistic effect has been described in the literature for the youngest age group which we could not analyse due to the broad and insufficient detail description of animal ages in the analysed studies. The results of our meta-analysis did not indicate a synergistic effect across all age groups, since the OR of a simultaneous infection of BRV and ETEC was not significant (OR: 0.94; 95% CI: 0.67–1.31; Figure 3).

Figure 3 indicates a synergistic effect between BRV and BCoV (OR:1.69; 95% CI: 1.32–2.16) and an inhibitory effect between BRV and Crypto (OR:0.77; 95% CI: 0.60–0.99). The former could be related to the fact that both BRV and BCoV increasingly cause diarrhoea in calves with failure of passive transfer [79], whereby weakening of the calf by one pathogen could also have a beneficial effect on other pathogens. In contrast to BRV, BCoV does not only infect the mature enterocytes in the small intestine but also the crypt cells and colonocytes [22,80]. The latter could be related to the fact that in the event of an infection with rotavirus endotoxin non-structural protein 4 (NSP4) is produced intracellularly and the upregulation of Ca^2+^ has an influence on the Ca^2+^-sensitive proteins F-actin, villin, and tubulin, resulting in damage of the microvillar cytoskeleton of the cell [77,81,82]. This or a similar pathophysiological effect might have an influence on the Crypto-binding capacity on the cell damaged by BRV. For example, Chen and colleagues described a decrease in infection of up to 70% with *Cryptosporidium parvum* induced by 2-actin depolymerisation in a vivo experiment [83]. This does not apply to a reverse appearance of infection as Tzipori and colleagues showed in lambs, where a previous infection with Crypto had no effect on BRV [84].

It is desirable to analyse prevalence data from numerous studies within the meta-analysis as it provides a more general overview of the influencing factors across the literature and countries. Thus, results of the meta-analysis are more powerful and less biased than conventional statistical methods and/or results of an individual study regarding NCD prevalences [13,38,85]. However, our study indicates the need for more standardised epidemiological studies to provide more robust conclusions regarding the importance of pathogens and their influencing factors. For instance, knowledge about the impact of other factors (e.g., vaccination of dams [30,55], supplementation of colostrum in the first two weeks of animal life [86,87], calf housing (place and individual vs. group pens) [28,29], routinely disposal and cleaning of bedding [26] as well as cleaning of feeding utensils [88] or quarantine of purchased animals [48]) on prevalence could help the livestock owner to reduce the direct production losses caused by NCD. The analysis of the effectiveness of specific measures against pathogens for which no vaccines are available yet (e.g., Crypto) would be essential to reduce the spread of zoonotic pathogens (in particular to minimise the health risk to farmers) and/or to reduce the use of drugs (in particular antibiotics). The authors also recommend taking into account the genotyping of the pathogens to identify possible mutations, to reassess if the vaccination strains match the field strains and to increase the understanding of the transmission of (zoonotic) pathogens. N.B. BRV and Crypto are important pathogens of diarrhoea in children and immunocompromised adults [89,90]. To determine the impact of potential influencing factors on the level of reported prevalences incurred by BRV-BCoV, BRV-ETEC, BRV-Crypto infections, it is desirable to have detailed and additional information on the prevalence, pathogens and animals, as shown in the study presented here.

## 5. Conclusions

As far as the authors are aware, this is the first worldwide meta-analysis to be carried out regarding the mixed infections of NCD. The results presented here revealed (i) a wide variation in the prevalence of the considered concurrent infections. The global prevalence of BRV-Crypto in calves (6.9%) was twice as high compared to that of BRV-BCoV (2.84%) and four times higher than BRV-ETEC (1.64%); (ii) calves with diarrhoea and in the age classes of sampled animals between 0–14 days showed the highest worldwide prevalence; iii) the chance to detect BCoV in calves with diarrhoea was higher in the presence of BRV compared to calves without BRV, whereby an inhibition effect was determined between BRV and Crypto infections; iv) diagnostic methods were identified as a significant influencing factor in detecting the considered mixed infections, while other factors differ related to their significance and explained variance on prevalences.

## Figures and Tables

**Figure 1 animals-11-01014-f001:**
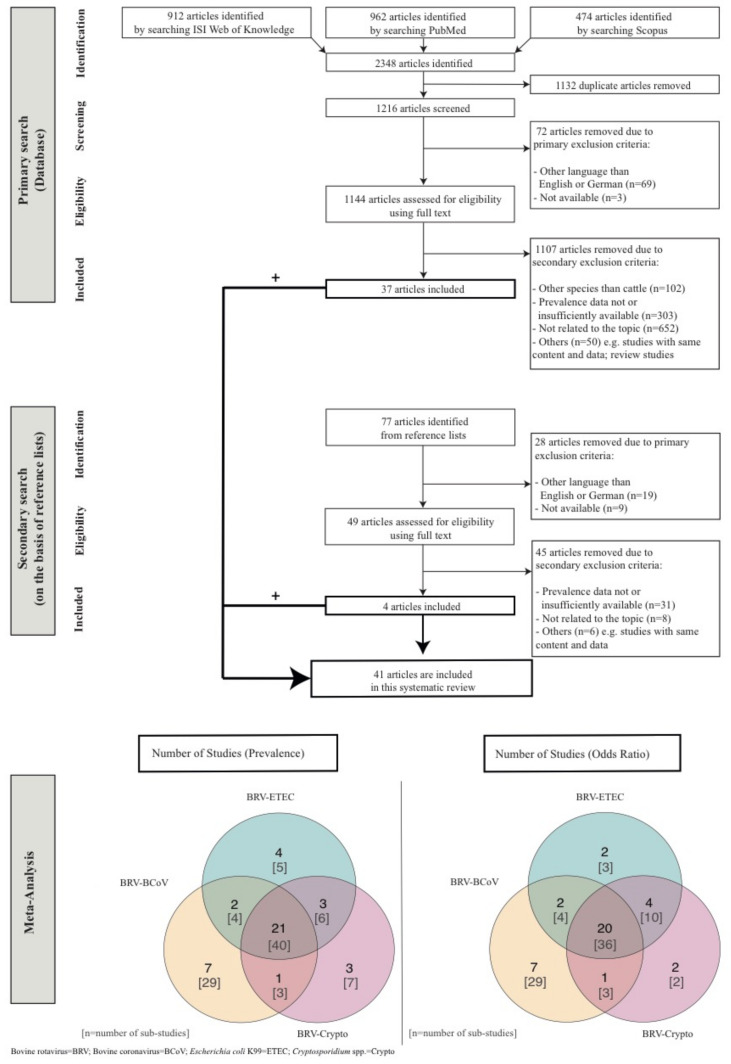
Flow chart of studies incorporated in the systematic review and meta-analysis.

**Figure 2 animals-11-01014-f002:**
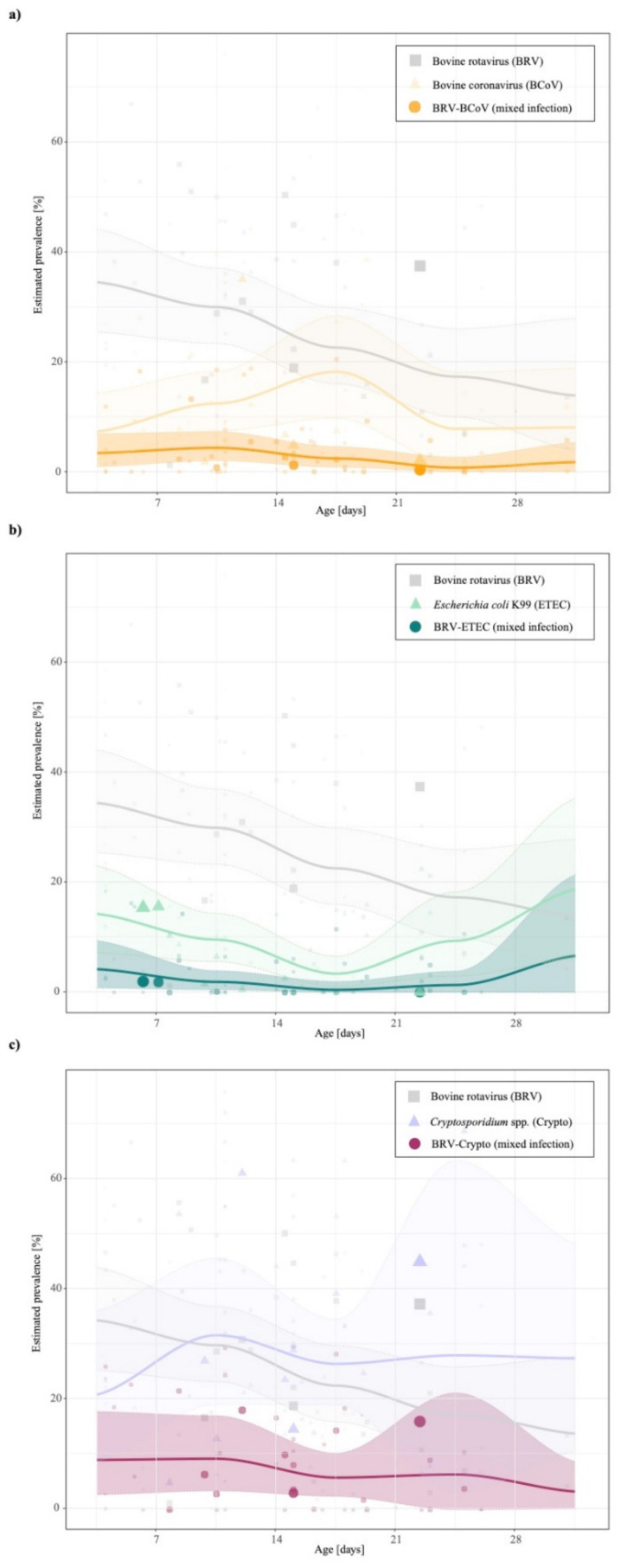
Temporal analysis of the individual (**a**) grey: bovine rotavirus; (**b**) orange: bovine coronavirus; (**c**) green: *Escherichia coli* K99; purple: *Cryptosporidium* spp.) and mixed prevalences stratified by age class of sampled animals until 30 days. The lines represent the mean prevalence estimates of all considered studies with the corresponding 95% CI (area) and individual prevalence points of studies (dots) during the period observed. The more prevalence estimates available at a certain age class of sampled animals, the wider the dots. N.B. To avoid imprecise model predictions, studies in the age groups (28–49 days) were excluded from the temporal curve fitting due to the small number of available studies (see number of available studies in Table 2, Table 3 and Table 4).

**Figure 3 animals-11-01014-f003:**
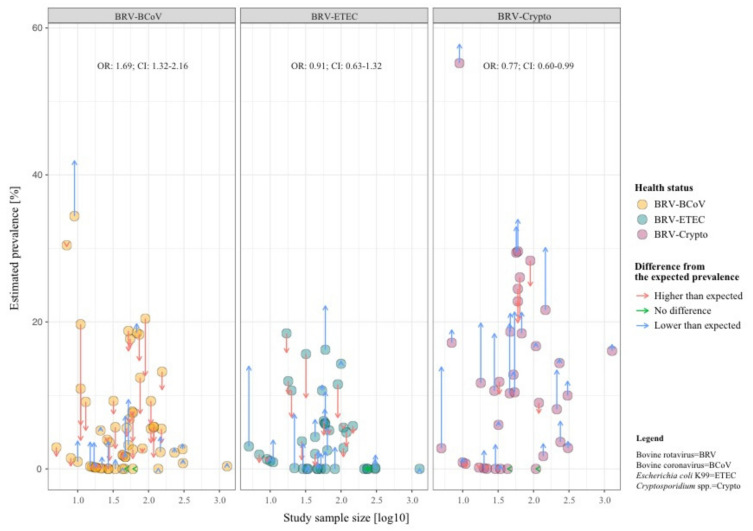
Comparison of the detected prevalence of mixed infections (dots) and the expected prevalence of infections (arrowheads) in calves with diarrhoea under the assumption of independency between both pathogens (e.g., P (BRV ∩ BCoV) = P(BRV) × P(BCoV)). Dots with blue arrows represent data with a lower prevalence than we would expect in an independent co-infection, while dots with red arrows represent data with a higher prevalence, as expected.

**Table 1 animals-11-01014-t001:** Collected data of the neonatal calf diarrhoea prevalence studies and analysed criteria in the meta-analysis and multivariate regression analysis.

Category	Systematic Review	(Subgroup) Meta-Analysis and Uni-vs. Multivariate Regression Analysis
Geographical region	Countries described the area where calves were tested.	Individual countries were assigned in respective regions (Europe, Australia, West Asia, East Asia, South Asia, North America, South America and Africa). The regions were included ^1^.
Sampling period	The date of sampling was defined as period begin and end of sampling. If the date of sampling was not mentioned, it was assigned to the category “not specified” and the submission date or publication date was used.	The sampling period were summarised in five time periods (1978–1980, 1981–1990, 1991–2000, 2001–2010, 2011–2019) and were included. The publication year and sampling period deviate from each other on average by three years. The period was included.
Number of herds	Only studies with greater or equal two herds were considered. The number of herds was recorded. If the number of herds was not mentioned, but it was described that several herds were sampled, it was assigned to the category “not specified”.	The number of herds was included in the meta-regression analysis.
Herd type	Herd types were categorised into dairy, beef and mixed (i.e., mixed covered more than one herd type). If the herd type was not mentioned, it was assigned to the category “not specified”.	The herd types (i.e., dairy, beef, mixed, not specified) were included.
Age (days)	Only calves under 60 days of age were considered. If the age was not mentioned but the animals were classified as “calves”, “neonatal” or “pre-weaned calves” the calves were assigned to the category “not specified”.	Age and age groups were very inconsistent across the different studies. In cases with published age median or mean we used that value, otherwise we calculated the center of the published age range. These centered data were clustered in 7-day periods. The seven-age class were included and ranged from 0–7 to 42–49 days.
Health status	Health status were categorised into diarrhoea, normal, mixed (i.e., mixed include both diarrhoea and normal). If the health status was not mentioned, it was assigned to the category “not specified”.	The health status (i.e., diarrhoea, normal, mixed, not specified) were included.
Samples size	Number of tested samples i.e., one sample per animal was included.	The number of tested samples were included in the subgroup meta-analysis and the meta-regression analysis.
Sample type	Sample types were categorised into autopsy, fecal and both (i.e., both covered more than one sample type).	The three sample types (i.e., autopsy, fecal, both) were included.
Prevalence of single and concurrent infections	All combinations of bovine rotavirus, bovine coronavirus, *Escherichia coli* K99 and *Cryptosporidium* spp. were considered as concurrent infections. Prevalence data, i.e., percentage and/or total number of tested and positively tested samples were recorded (including individual for bovine rotavirus, bovine coronavirus, *Escherichia coli* K99 and *Cryptosporidium* spp.) If only percentage data were available, then the number of positive samples was extrapolated.	Only combinations of bovine rotavirus with other pathogens were included. Cumulative and absolute numbers of BRV-BCoV, BRV-ETEC and BRV-Crypto were included (i.e., including triple and quadruple infections). If a combination of these pathogens was not present it was considered as zero prevalence to avoid publication bias. The latter would have happened if only positive combinations were considered.
Genotype	Information of *Cryptosporidium* spp. was collected.	The genotypes were not incorporated in the meta-analysis due to insufficient number of studies and data.
Vaccination status	The vaccination status of the dam was collected with “Yes” or “No”. If only a part of the tested herds were vaccinated, then it was assigned to the category “partly vaccinated”. If the vaccination status was not mentioned, it was assigned to the category “not specified”.	The vaccination status was not incorporated in the meta-analysis due to insufficient data. More than 75% of the studies not specified the vaccination status of the dam.
Colostrum	Assurance of colostrum intake was assigned to the category “Yes” or “No”. If failure of passive transfer of maternal antibodies was diagnosed it was categorised as “deficient”. If the assurance of colostrum intake was not mentioned, it was assigned to the category “not specified”.	The assurance of colostrum intake was not incorporated in the meta-analysis due to insufficient data. More than 90% of the studies did not report data of the colostrum intake (i.e., Ig/L or TP/L).
Diagnostic method	Types of diagnostic methods (e.g., PCR^a^, ELISA^b^, acid-fast staining) used were collected and wherever available the corresponding sensitivity and specificity were recorded. If the diagnostic method was not mentioned, it was assigned to the category “not specified”.	The different applied diagnostic methods were classified per pathogen as follows: -Diagnostic method (BRV) covered PCR ^a^, ELISA ^b^, RA ^c^ and EM ^d^ as single detection method; “Several” covered combinations of diagnostic methods for screening and confirmation of laboratory results (parallel interpretation of tests) as follows: EM ^d^, ELISA ^b^, IF ^h^, LA ^o^, PAGE ^f^, RA ^b^, FAT ^i^, PCR ^a^; “other im” covered other antibody-based methods besides ELISA ^b^ and RA ^c^ (i.e., IHC ^j^, FAT ^i^). -Diagnostic method (BCoV) covered PCR ^a^, ELISA ^b^, RA ^c^ and EM ^d^ as single detection method; “Several” covered combinations of diagnostic methods for screening and confirmation of laboratory results (parallel interpretation of tests) as follows: ELISA ^b^, FAT ^i^, IF ^h^, EM ^d^, SPIEM ^e^, HEHA ^k^, HE ^l^, PCR ^a^); “other im” covered other antibody-based methods besides ELISA ^b^ and RA ^c^ (i.e., IF ^h^, IHC ^j^, HEHA ^k^, HAI ^m^). -Diagnostic method (ETEC) covered agglutination (i.e., bacterial culture followed by SA ^n^ or LA ^o^), ELISA ^b^ and RA ^c^ as single detection method; “Several” covered combinations of diagnostic methods for screening and confirmation of laboratory results (parallel interpretation of tests) as follows: ELISA ^b^, IF ^h^, IHC ^j^, RA ^c^, Agglutination, PCR ^a^. -Diagnostic method (Crypto) covered MS ^g^, ELISA ^b^ and RA ^c^ as single detection method; “Several” covered combinations of diagnostic methods for screening and confirmation of laboratory results (parallel interpretation of tests) as follows: MS ^g^, ELISA ^b^, RA ^c^, IF ^h^, PCR ^a^. The diagnostic methods were included. The prevalence was not corrected to the test sensitivity and specificity due to insufficient data. More than 75% of the studies not provided the information.
Study type	The studies were categorised into three levels. (1) Case-control study: studies tested diarrheic and normal calves and/or sampling were performed in several regions of a country; (2) Case study: studies tested only cases with diarrhoea and/or testing were performed in several regions of a country; (3) Other studies: studies focusing on diagnostic of pathogens.	The three study types were included.

^1^ = The term “were included” defined, that the factor was included in the overall- and subgroup meta-regression analysis as well as in the meta-regression analysis. ^a^ Polymerase chain reaction, ^b^ Antigen enzyme-linked immunosorbent assay, ^c^ Lateral flow immunochromatographic assay, ^d^ Electron microscopy, ^e^ Solid-phase immuno electron microscopy, ^f^ Polyacrylamide gel electrophoresis, ^g^ Acid-fast staining, ^h^ Immunofluorescence assay, ^i^ Fluorescence antibody technique, ^j^ Immunohistochemical/immunostaining, ^k^ Hemadsorption-elution-hemagglutination assay, ^l^ Hemagglutination-elution assay, ^m^ Hemagglutination-inhibition assay, ^n^ Slide agglutination, ^o^ Latex agglutination.

**Table 2 animals-11-01014-t002:** Subgroup meta-analysis of studies reporting the concurrent prevalence of bovine rotavirus (BRV) and bovine coronavirus (BCoV). N.B. Detailed description of the factors is provided in Table 1.

BRV-BCoV							
	Sample Size (No. Animals)	No. Studies	No. Prevalence Inputs	Weighted Mean Estimate (%)	Confidence Interval (95%)	Qep ^a^	I^2^ (%) ^b^
Overall	6974	31	76	2.84	(1.78–4.08)	<0.01	77.97
Geographical region							
Europe	3841	15	30	4.72	(2.49–7.45)	<0.01	87.61
North America	487	5	10	2.63	(0.19–6.79)	<0.01	71.28
South America	-	-	-	-	-	-	-
East Asia	251	1	2	0.00	(0.00–0.07)	0.95	0.00
West Asia	366	3	8	2.35	(0.50–5.09)	0.18	27.92
South Asia	393	3	17	1.19	(0.13–2.90)	0.68	0.00
Oceania	1226	1	2	1.22	(0.51–2.15)	0.91	0.00
Africa	410	3	7	2.72	(0.21–6.97)	<0.01	62.42
Period							
1978–1980	59	1	2	3.16	(0.00–17.62)	0.07	69.83
1981–1990	1437	7	19	5.48	(2.32–9.58)	<0.01	82.59
1991–2000	1177	7	17	4.64	(2.35–7.48)	<0.01	68.22
2001–2010	1197	9	22	0.86	(0.02–2.48)	<0.01	61.55
2011–2019	3104	7	16	1.54	(0.49–2.98)	0.01	61.18
Herd type							
Dairy	3057	16	37	3.44	(1.91–5.28)	<0.01	72.78
Beef	91	1	3	0.62	(0.00–4.27)	0.72	0.00
Mixed	832	4	8	0.78	(0.00–2.57)	0.10	45.81
Not specified	2994	10	28	3.17	(1.21–5.75)	<0.01	84.18
Age class (in days)							
0–7	926	11	14	3.39	(0.91–6.92)	<0.01	70.06
7–14	1615	16	23	4.35	(2.02–7.30)	<0.01	78.18
14–21	2314	16	22	2.43	(0.82–4.60)	<0.01	77.19
21–28	1901	9	11	0.74	(0.00–2.68)	<0.01	68.42
28–35	207	4	5	1.75	(<0.01–5.34)	0.19	20.13
35–42	-	-	-	-	-	-	-
42–49	11	1	1	19.68	(1.90–47.48)	1.00	0.00
Health status							
Diarrhoea	4975	29	59	4.22	(2.83–5.82)	<0.01	75.29
Normal	577	11	14	0.00	(0.00–0.25)	0.99	0.00
Mixed	196	1	1	0.00	(0.00–0.06)	1.00	0.00
Not specified	1226	1	2	1.22	(0.51–2.15)	0.91	0.00
Sample type							
Fecal	6437	1	67	2.18	(1.27–3.26)	<0.01	73.20
Autopsy	457	28	4	11.57	(6.68–17.42)	0.03	64.00
Both	80	2	5	10.55	(0.82–27.00)	0.02	66.04
Diagnostic method (BRV)							
ELISA	1183	9	25	1.62	(0.35–3.48)	<0.01	59.40
RA ^c^	1576	4	8	1.80	(0.23–4.28)	0.03	48.79
Several	3684	15	33	3.31	(1.84–5.09)	<0.01	76.75
EM ^d^	259	1	5	9.50	(3.80–17.02)	0.02	66.17
Other im ^e^	21	1	3	22.50	(5.00–46.74)	0.25	29.86
PCR	251	1	2	0.00	(0.00–0.07)	0.95	0
Diagnostic method (BCoV)							
ELISA	2735	12	32	1.95	(0.77–3.49)	<0.01	67.89
RA	1576	4	8	1.80	(0.23–4.28)	0.03	48.79
Several	1424	8	13	1.46	(0.42–2.92)	0.07	48.03
EM	418	3	9	7.05	(2.52–13.16)	<0.01	73.18
Other im	570	3	12	9.85	(5.68–14.84)	0.04	52.84
PCR	251	1	2	0.00	(0.00–0.07)	0.95	0
Study type							
Case-control	3486	13	34	0.67	(0.16–1.40)	<0.01	46.56
Case	3368	16	40	5.48	(3.55–7.71)	<0.01	74.36
Other	120	2	2	9.84	(0.06–28.94)	0.01	85.43

^a^ Qep = The Q statistic and its p-value serve as a test of significance [43]. ^b^ I^2^ = The ratio of true heterogeneity to total variation in observed effects [43]. ^c^ RA = Lateral flow immunochromatographic assay. ^d^ EM = Electron microscopy. ^e^ Other im = Other antibody based methods besides ELISA and RA (i.e., IHC, FAT).

**Table 3 animals-11-01014-t003:** Subgroup meta-analysis of studies reporting the concurrent prevalence of bovine rotavirus (BRV) and *Escherichia coli* K99 (ETEC) N.B. Detailed description of the factors is provided in Table 1.

BRV-ETEC							
	Sample Size (No. Animals)	No. Studies	No. Prevalence Inputs	Weighted Mean Estimate (%)	Confidence Interval (95%)	Qep ^a^	I^2^ (%) ^b^
Overall	8897	30	55	1.64	(0.76–2.75)	<0.01	83.88
Geographical region							
Europe	6692	17	27	0.97	(0.17–2.20)	<0.01	85.09
North America	326	4	8	3.62	(0.50–8.57)	<0.01	64.32
South America	663	2	4	0.15	(0.00–3.32)	<0.01	86.51
East Asia	-	-	-	-	-	-	-
West Asia	366	3	8	3.70	(0.32–9.39)	<0.01	74.42
South Asia	93	1	4	3.40	(0.00–11.80)	0.14	49.67
Oceania	429	1	1	1.20	(0.17–2.82)	1.00	0.00
Africa	328	2	3	2.43	(0.30–5.85)	0.14	49.22
Period							
1978–1980	159	2	3	8.36	(0.21–23.56)	0.01	81.53
1981–1990	4955	9	17	0.54	(0.00–1.74)	<0.01	82.68
1991–2000	820	6	12	1.22	(0.11–3.08)	<0.01	50.89
2001–2010	738	7	12	1.57	(0.04–4.37)	<0.01	68.10
2011–2019	2225	6	11	3.08	(0.62–6.76)	<0.01	87.15
Herd type							
Dairy	2556	16	27	1.90	(0.76–3.39)	<0.01	70.01
Beef	304	2	4	0.00	(0.00–0.12)	0.82	0.00
Mixed	895	5	10	0.11	(0.00–1.54)	0.02	61.71
Not specified	5142	8	14	3.75	(1.18–7.30)	<0.01	93.38
Age class (in days)							
0–7	2495	8	9	4.24	(0.81–9.45)	<0.01	89.03
7–14	2769	13	18	1.92	(0.49–3.96)	<0.01	78.32
14–21	1559	12	16	0.48	(0.00–2.00)	<0.01	73.83
21–28	2035	7	9	1.36	(0.02–3.87)	<0.01	81.80
28–35	28	2	2	6.67	(0.00–21.5)	0.26	20.31
35–42	-	-	-	-	-	-	
42–49	11	1	1	0.91	(0.00–16.82)	1.00	0.00
Health status							
Diarrhoea	7509	28	42	2.26	(1.04–3.79)	<0.01	87.27
Normal	763	1	11	0.13	(0.00–0.80)	0.66	0.00
Mixed	196	1	1	0.00	(0.00–1.18)	1.00	0.00
Not specified	429	1	1	1.20	(0.17–2.82)	1.00	0.00
Sample type							
Fecal	5624	26	48	1.54	(0.62–2.72)	<0.01	80.75
Autopsy	-	-	-	-	-	-	-
Both	3273	4	7	2.52	(0.62–5.28)	0.22	74.52
Diagnostic method (BRV)							
ELISA	1272	9	16	3.01	(0.95–5.85)	<0.01	75.94
RA ^c^	1576	4	8	3.90	(0.43–9.57)	<0.01	83.39
Several	2835	14	26	0.59	(0.02–1.66)	<0.01	71.85
Other im ^d^	21	1	3	1.92	(0.00–13.11)	0.98	0.00
Not specified	3193	2	2	1.95	(1.39–2.58)	0.84	0.00
Diagnostic method (ETEC)							
ELISA	183	2	5	5.36	(0.60–13.18)	0.04	59.51
RA	1576	4	8	3.90	(0.43–9.57)	<0.01	83.39
Several	1426	10	18	1.58	(0.34–3.40)	<0.01	61.00
Agglutination ^e^	5712	14	24	0.85	(0.09–2.09)	<0.01	85.43
Study type							
Case-control	2622	13	27	1.66	(0.45–3.35)	<0.01	78.15
Case	6223	16	27	1.77	(0.56–3.44)	<0.01	86.91
Other	52	1	1	0.00	(0.00–2.82)	1.00	0.00

^a^ Qep = The Q statistic and its *p*-value serve as a test of significance [43]. ^b^ I^2^ = The ratio of true heterogeneity to total variation in observed effects [43]. ^c^ RA = Lateral flow immunochromatographic assay. ^d^ Other im = Other antibody based methods besides ELISA and RA (i.e., IHC, FAT). ^e^ Agglutination = Bacterial culture followed by slide agglutination or latex agglutination.

**Table 4 animals-11-01014-t004:** Subgroup meta-analysis of studies reporting the concurrent prevalence of bovine rotavirus (BRV) and *Cryptosporidium* spp. (Crypto) N.B. Detailed description of the factors is provided in Table 1.

BRV-Crypto							
	Sample Size (No. Animals)	No. Studies	No. Prevalence Inputs	Weighted Mean Estimate (%)	Confidence Interval (95%)	Qep ^a^	I^2^ (%) ^b^
Overall	7191	28	56	6.69	(4.27–9.51)	<0.01	92.55
Geographical region							
Europe	4235	16	26	8.90	(4.98–13.65)	<0.01	94.03
North America	240	3	6	6.59	(0.21–18.16)	<0.01	83.92
South America	452	1	2	5.68	(1.99–10.67)	0.07	69.09
East Asia	251	1	2	0.79	(0.00–3.08)	0.28	15.43
West Asia	266	2	6	16.61	(8.03–27.19)	0.01	73.82
South Asia	193	2	9	1.98	(0.00–6.76)	0.03	48.39
Oceania	1226	1	2	2.27	(0.91–4.05)	0.14	52.98
Africa	328	2	3	0.62	(0.00–8.81)	<0.01	91.29
Period							
1978–1980	59	1	2	2.04	(0.00–11.65)	0.16	49.67
1981–1990	1465	5	11	3.92	(0.10–10.93)	<0.01	94.44
1991–2000	820	6	12	8.46	(2.66–16.47)	<0.01	89.89
2001–2010	1313	8	13	5.62	(1.79–10.88)	<0.01	88.07
2011–2019	3534	8	18	9.07	(4.72–14.44)	<0.01	93.35
Herd type							
Dairy	3892	16	29	6.13	(2.90–10.19)	<0.01	93.46
Beef	304	2	4	5.64	(0.03–16.64)	0.01	80.25
Mixed	895	5	10	3.23	(0.61–7.17)	<0.01	75.81
Not specified	2100	6	13	12.37	(6.04–20.24)	<0.01	91.71
Age class (in days)							
0–7	690	8	9	9.04	(2.72–17.87)	<0.01	83.75
7–14	1943	12	15	9.27	(3.42–17.10)	<0.01	94.83
14–21	2637	15	20	5.84	(2.41–10.29)	<0.01	92.00
21–28	1843	7	8	3.28	(0.16–8.74)	<0.01	90.48
28–35	28	2	2	6.41	(0.00–21.29)	0.26	20.31
35–42	-	-	-	-	-	-	-
42–49	50	2	2	7.80	(0.00–27.97)	0.11	60.99
Health status							
Diarrhoea	4269	24	42	9.43	(6.28–13.06)	<0.01	89.62
Normal	664	9	9	0.00	(0.00–0.03)	0.60	0.00
Mixed	1032	3	3	8.78	(2.19–18.29)	<0.01	94.30
Not specified	1226	1	2	2.27	(0.91–4.05)	0.14	52.98
Sample type							
Fecal	7111	26	51	6.50	(4.09–9.32)	<0.01	92.72
Autopsy	-	-	-	-	-	-	-
Both	80	2	5	10.96	(0.04–32.23)	<0.01	77.94
Diagnostic method (BRV)							
ELISA	1086	7	16	2.21	(0.06–6.16)	<0.01	84.92
RA ^c^	2412	6	10	14.94	(9.76–20.89)	<0.01	87.27
Several	3421	13	25	7.01	(3.81–10.90)	<0.01	91.33
Other im ^d^	21	1	3	23.33	(0.97–59.14)	0.05	65.92
PCR	251	1	2	0.79	(0.00–3.08)	0.28	15.43
Diagnostic method (Crypto)							
ELISA	93	1	4	<0.01	(0.00–2.39)	0.97	0.00
RA	1794	4	4	13.49	(6.80–21.74)	<0.01	89.72
Several	2957	11	24	12.21	(7.27–15.75)	<0.01	89.01
MS ^e^	2347	12	24	3.44	(0.85–7.16)	<0.01	91.00
Study type							
Case-control	4109	13	26	4.27	(1.89–7.30)	<0.01	91.55
Case	3030	14	29	9.29	(5.13–14.34)	<0.01	91.39
Other	52	1	1	12.82	(4.59–23.81)	1.00	0.00

^a^ Qep = The Q statistic and its *p*-value serve as a test of significance [43]. ^b^ I^2^ = The ratio of true heterogeneity to total variation in observed effects [43]. ^c^ RA = Lateral flow immunochromatographic assay. ^d^ Other im = Other antibody based methods besides ELISA and RA (i.e., IHC, FAT). ^e^ MS = acid-fast staining.

**Table 5 animals-11-01014-t005:** Uni- and multivariate meta-regression analysis stratified by factors and type of double mixed infection.

Univariate (BRV-BCoV)			Univariate (BRV-ETEC)			Univariate (BRV-Crypto)			
Factors	R^2^	*p*	Factors	R^2^	*p*	Factors	R^2^	*p*	
Region	7.68	0.12 *	Region	0.14	0.42	Region	12.01	0.07 *	
Period	13.78	0.02 *	Period	11.42	0.07 **	Period	0.00	0.53	
Number of herds	0.00	0.81	Number of herds	10.65	0.03 **	Number of herds	0.00	0.57	
Herd type	1.11	0.29	Herd type	15.60	0.02 **	Herd type	4.32	0.17 *	
Age class	4.86	0.17 *	Age class	2.13	0.34	Age class	0.00	0.54	
Health status	30.25	0.00 **	Health status	2.28	0.27	Health status	27.32	<0.01 **	
Sample size	0.00	0.30	Sample size	0.71	0.26	Sample size	0.00	0.69	
Sample type	20.66	<0.01 *	Sample type	0.00	0.48	Sample type	0.00	0.56	
Diagnostic BRV	27.22	<0.01 **	Diagnostic BRV	5.25	0.20 *	Diagnostic BRV	22.81	<0.01 *	
Diagnostic BCoV	38.20	0.00 **	Diagnostic ETEC	6.32	0.14 *	Diagnostic Crypto	22.51	<0.01 **	
Study type	39.31	0.00 **	Study type	0.00	0.67	Study type	4.51	0.13 *	
**Multivariate (BRV-BCoV)**			**Multivariate (BRV-ETEC)**			**Multivariate (BRV-Crypto)**			
Number of factors	R^2^	AICc/*p* Value LRT	Number of factors	R^2^	AICc/*p* Value LRT	Number of factors	R^2^		AICc/*p* Value LRT
Full Model (n = 8; *p* < 0.25 *)	61.23	−63.27/-	Full Model (n = 5; *p* < 0.25 *)	47.83	−60.11/-	Full Model (n = 6; *p* < 0.25 *)	46.20		−4.57/-
Reduced Model (n=4 **)	59.75	−114.01/0.07	Reduced Model (n = 3 **)	37.82	−71.64/0.03	Reduced Model (n = 2 **)	49.54		−48.71/0.05

R^2^ = Coefficient of determination; R-squared is a goodness-of-fit measure for regression models and indicates the proportion of variance in the dependent variable that can be explained by the independent variable. AICc = Akaike information criterion, corrected for small sample size. *p* Value LRT = The *p* value (significance level) of the likelihood ratio test (LRT). * = Significant factors in the full model. ** = Significant factors in the reduced model

## Data Availability

The data presented in this study are available in the supplementary material. The digital datasets and analyses of the present study are available from the corresponding author upon request.

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
