# Peer review of "Prevalence of Worldwide Neonatal Calf Diarrhoea Caused by Bovine Rotavirus in Combination with Bovine Coronavirus, Escherichia coli K99 and Cryptosporidium spp.: A Meta-Analysis"

_animals, 2021, doi:10.3390/ani11041014_

Round 1

Reviewer 1 Report

The submitted manuscript presents the results of three weighted-stratified random-effects meta-analyses performed to calculate the worldwide prevalence of mixed infections of the causative agents of neonatal calf diarrhoea: Bovine Rotavirus-Bovine Coronavirus, Bovine Rotavirus -Enterotoxigenic Escherichi coli, and Bovine Rotavirus -Cryptosporidium spp, and their potential influencing factors. The meta-analysis included 41 studies in 21 countries where the presence or absence of mixed infections in global calf populations has been determined. The highest worldwide estimated pooled prevalence identified for Bovine Rotavirus-Cryptosporidium spp was 6.69%, followed by 2.84% and 1.64% for Bovine Rotavirus-Bovine Coronavirus, and Bovine Rotavirus -Enterotoxigenic Escherichia coli mixed infections, respectively

The study presented is the first worldwide meta-analysis to be carried out regarding the mixed infections of neonatal calf diarrhoea. The manuscript is well written, the meta-analyses are adequately designed and the results obtained are statistically well substantiated, indicating that diagnostic methods were identified as a significant influencing factor in detecting the considered mixed infections, while other factors differ related to their significance and explained variance on prevalences. One of the probable influencing factors that is not dealt with in the manuscript is the role of the gut microbiota during pathogenesis at the site of infection in early life, and the understanding of the host-microbial interactions with dietary interventions. This is understandable, as these studies have been published relatively very recently. However it is advisable to invite authors to ellaborate a short paragraph on this topic in the discusion section of the manuscript based in the results presented in some of the following references:

Malmuthuge, N., Griebel P.J., Guan L. L. The gut microbiome and its potential role in the development and function of newborn calf gastrointestinal tract. Frontiers in Veterinary Science. 2, 2015, 36. https://www.frontiersin.org/article/10.3389/fvets.2015.00036          

Malmuthuge, N., Guan L.L. Understanding the gut microbiome of dairy calves: Opportunities to improve early-life gut health. Journal of Dairy Science. 100 (7). 2017. 5996-6005. https://doi.org/10.3168/jds.2016-12239.

Alipour, M.J., Jalanka, J., Pessa-Morikawa, T., Kokkonen, T., Satokari, R., Hynonen U., Livanainen, A., Niku, M. The composition of the perinatal intestinal microbiota in cattle. Sci Rep 8, 10437 (2018). https://doi.org/10.1038/s41598-018-28733-y

Osorio, J.S. Gut health, stress, and immunity in neonatal dairy calves: the host side of host-pathogen interactions. J Animal Sci Biotechnol 11, 105 (2020). https://doi.org/10.1186/s40104-020-00509-3

Hang, B.P.T., Wredle, E., Dicksved, J. Analysis of the developing gut microbiota in young dairy calves—impact of colostrum microbiota and gut disturbances. Trop Anim Health Prod 53, 50 (2021). https://doi.org/10.1007/s11250-020-02535-9

Author Response

Thank you for this comment. We have now incorporated all suggested references in the discussion section as follows (please on page 19, lines 396-400; new text underlined): “One of the influencing factors that was also not considered in the study presented here is the role of the gut microbiome during pathogenesis at the site of infection in the early life of an animal, and the host-microbial interactions with dietary interventions. A number of studies have analysed the effect of the microbiome composition on new-born health such as on the calf gastrointestinal tract [61–65].”

Reviewer 2 Report

The authors present the results of three weighted-stratified random-effects meta-analyses aimed to estimate the worldwide prevalence of neonatal calf diarrhoea of mixed infections of some of the most common causative agents. In particular they consider data about concurrent infections of bovine rotavirus and bovine coronavirus, or Escherichia coli K99 or Cryptosporidium spp.

The work is appropriately conducted and presented.

I ask the authors to consider some revisions and improvements as outlined below.

  • At lines 58-89 329-330 and 475-476 the authors underline the originality of their work. Anyway, this is not “it-self” an important additional value for the work. It would be more important that they would provide a reason of the significance of their novel data. They should better explain why it is valuable to estimate the worldwide prevalence of mixed infection of neonatal calf.
  • In the introduction paragraph (lines 50-52) the authors state that “Bovine rotavirus (BRV), bovine coronavirus (BCoV), enterotoxigenic Escherichia coli K99 (ETEC) and Cryptosporidium parvum are the most frequently identified causative factors of calf diarrhoea during the first 30 days of age”. However, in the “material and methods” paragraph, at lines 111-112, the authors affirm that they include in the study data from calf of less than 60 day of age. Moreover, in table 1, page 5, the authors state that age data were divided in classes of 7 days from 0 to 49 days and finally in Figure 2 in temporal analysis of individual and mixed prevalences they consider the stratified classes until 30 days (28?). Please clarify.
  • At line 108, the Excel version should be indicated.

Author Response

Thank you for your helpful comment. We have now included the reason why its valuable to estimate the worldwide prevalence of mixed infection of neonatal calf in the discussion section as follows (on page 21, lines 466-470 new text underlined):

“It is desirable to analyse prevalence data from numerous studies within the meta-analysis as it provides a more general overview of the influencing factors across the literature and countries. Thus, results of the meta-analysis are more powerful and less biased than conventional statistical methods and/or results of an individual study regarding NCD prevalences [13,38,85].”

Thank you for your helpful comment. We have now incorporated for the reader of the journal the following explanation in the material section as follows (on page 3, lines 115-121; new text underlined): “Although NCD is the most common cause of death in dairy calves during their first 30 days of age, we considered studies with age ranges up to 60 days in the analysis because many of the studies published age ranges including animals older than 30 days. Further, age ranges up to 60 days were considered to get a better impression of the development of mixed infection in both dairy and beef production systems instead of single infection in one production system, and to confirm the knowledge in the literature that NCD most frequently appears in the first 30 days of age.“

In the results section, we have now included the following sentence (please see on page 9, lines 236-237; new test underlines): “Our study results confirm that the most concurrent infections occur in dairy and beef production systems in age range up to 30 days (see Table 2-4)”

Additionally, we have now included the following explanation in the caption of Figure 2 (please see on page 17, lines 321-323; new test underlined): N.B. To avoid imprecise model predictions, studies in the age groups (28-49 days) were excluded from the temporal curve fitting due to the small number of available studies (see number of available studies in Table 2-4).”

Done as suggested (please see on page 3, line 108 (new text underlined)): “…were entered into a Microsoft Excel datasheet Version 16.16.27 (2016)”
